# Functional Network Connectivity for Components of Depression-Related Psychological Fragility

**DOI:** 10.3390/brainsci14080845

**Published:** 2024-08-22

**Authors:** Ian D. Evans, Christopher F. Sharpley, Vicki Bitsika, Kirstan A. Vessey, Emmanuel Jesulola, Linda L. Agnew

**Affiliations:** 1Brain-Behaviour Research Group, School of Science & Technology, University of New England, Armidale, NSW 2351, Australia; ievans3@une.edu.au (I.D.E.); vicki.bitsika@une.edu.au (V.B.); kvessey@une.edu.au (K.A.V.); doctorseasept@yahoo.com (E.J.); linda.agnew@griffith.edu.au (L.L.A.); 2Department of Neurosurgery, The Alfred Hospital, Melbourne, VIC 3004, Australia; 3Griffith Health Group, Griffith University, Southport, QLD 4222, Australia

**Keywords:** fragility, resilience, depression, functional connectivity, default mode network, central executive network, salience network

## Abstract

Psychological resilience (PR) is known to be inversely associated with depression. While there is a growing body of research examining how depression alters activity across multiple functional neural networks, how differences in PR affect these networks is largely unexplored. This study examines the relationship between PR and functional connectivity in the alpha and beta bands within (and between) eighteen established cortical nodes in the default mode network, the central executive network, and the salience network. Resting-state EEG data from 99 adult participants (32 depressed, 67 non-depressed) were used to measure the correlation between the five factors of PR sourced from the Connor–Davidson Resilience Scale and eLORETA-based measures of coherence and phase synchronisation. Distinct functional connectivity patterns were seen across each resilience factor, with a notable absence of overlapping positive results across the depressed and non-depressed samples. These results indicate that depression may modulate how resilience is expressed in terms of fundamental neural activity.

## 1. Introduction

Although Major Depressive Disorder (MDD) is often referred to as a unitary construct, there are multiple research data which demonstrate that it is heterogeneous [1], based on various combinations of nine key symptoms and several associated features [2]. These combinations have been referred to as “depression subtypes”, and several of these subtypes have been defined. For example, the DSM-5-TR [2] lists MDD with anxiety, with mixed features, with melancholic features, and with atypical features, plus others. Other MDD subtypes have been described by various authors [3,4,5,6], including the possible different neurobiological pathways and circuits they may have. Not surprisingly, there are also suggestions that these depression subtypes may require a variety of treatments [7,8,9], and that different patient behaviour patterns may be differentially effective with different depression subtypes.

### 1.1. Resilience and Fragility

One such behaviour pattern that has been shown to help people who experience chronic or acute stress recover from the deleterious effects of that stress is “psychological resilience” (PR; [5]). PR has been defined as a set of behavioural skills that assist in successful adaptation to difficult or challenging life experiences, especially via mental, emotional, and behaviour flexibility and adjustment to external and internal demands [9]. A lack of resilience (i.e., fragility) has been observed in those with MDD, both in terms of behavioural analyses [10,11,12] and functional connectivity [13]. Additionally, biomarkers such as decreased glutamate response in the medial prefrontal cortex predicts an appropriate response to acute stress, while no change in this glutamatergic response is consistently found in those with MDD [14].

However, like MDD, PR is not unitary. Rather, it has been described as comprising multiple components, such as personal competence, high standards, tenacity; trust in one’s instincts, tolerance of negative affect, strengthening effects of stress; positive acceptance of change, secure relationships; control; and spiritual influences [15]. Not all of these components are equally effective in reducing depression [16], and it may therefore be hypothesised that, as well as representing different skill sets, they may also be aligned with different underlying neurological bases. Mapping these underlying neurological phenomena could assist in more efficient matching of PR components with specific MDD subtypes. Although several recent papers have reported on the association between various neurological phenomena (such as EEG networks) and PR, they have done so using PR as a unitary construct [17,18,19] rather than examining the possibly different neurological correlates of the different PR components. As a result, it may be the case that specific areas of psychological fragility can be associated with specific subtypes of depression rather than a broad, overarching concept of both resilience and depression.

### 1.2. Neural Networks and Functional Connectivity

EEG has a long history of use in observing the dynamics of functional networks in different psychiatric conditions, and multiple findings indicate atypical functional connectivity is commonly found in persons with MDD (e.g., [20,21]). The major networks typically examined when searching for neural correlates of depression are the default mode network (DMN [22,23,24,25], the central executive network (CEN [26,27]), and the salience network (SN [27,28]). In each case, connectivity is assessed either via MRI or EEG studies, with the requisite trade-off in spatial and temporal resolution that each neuroimaging technique offers. A balance between these limitations in spatial and temporal resolution can be achieved using techniques such as lagged coherence via eLORETA [29], which makes it possible to assess functional connectivity within known frequency bands (e.g., theta, alpha, beta) at specific cortical regions known to be associated with each of these networks.

Making predictions regarding which neural networks or frequency bands to study in advance is complicated by two factors before even considering emotional resilience: (a) inconsistent functional connectivity results from studies which have included depressed populations and (b) the scarcity of high-quality studies examining functional connectivity of neural networks in depressed populations. For example, the 2023 systematic review by Miljevic and colleagues [30] covering 52 functional connectivity studies focusing on depression found that functional connectivity across any frequency band in depressed populations could be higher, lower, or not significantly different in either direction to the general population, leading the authors to conclude (p. 287, [30]) “while most resting-state studies noted a difference in alpha, theta, and beta, no clear conclusions could be drawn about the direction of the difference”.

The methodology concern is compounded by the lack of functional network studies. EEG studies examining neural networks in depressed people are prone to analysing data at a range of electrode pairings (e.g., [25]) rather than source estimates of the relevant deeper cortical structures using methods such as eLORETA. While there are some studies that examine functional connectivity differences in depressed people using eLORETA (e.g., [31,32]), no such studies have focused on known functional neural networks.

Therefore, to address this gap in the literature, and to provide some greater understanding of how a lack of PR and depression interact within select functional neural networks, this study used eLORETA to examine the effect of PR on eyes-closed resting-state theta, alpha, and beta EEG coherence within and between major nodes of the DMN, CEN, and SN. The effect of PR on coherence was contrasted between depressed and non-depressed samples.

## 2. Materials and Methods

Statistical power analysis indicated that a minimum sample of 71 was required, assuming α-error probability of 0.1, power of 0.9, and H_1_ correlations of at least 0.3. A total sample of 106 adults aged 18 years old or more from the New England region of New South Wales, Australia, were recruited from a media-advertised study investigating mental health in rural and regional communities of Australia as part of the New England Mental Health Study. Participants were screened for the following: no previous medical history of severe physical brain injury, previous brain surgery, past or current history of epilepsy or seizure disorder, claustrophobia (EEG data were collected in a small booth), or undergoing pharmacotherapy at the time of data collection. Six participants were excluded based on these criteria. One further participant was excluded from the EEG analysis due to an exceptionally low number of artefact-free epochs in the baseline condition, resulting in a final sample of 99 (46 females).

The Connor–Davidson Resilience Scale (CDRISC; [15]) comprises 25 items such as “I like a challenge”, “When things look hopeless I don’t give up”, “I bounce back after illness or hardship”, and “I am able to adapt to change”. Each of these items is rated by respondents by the selection of one score on a 5-point scale for how true the statement was for them during the last month (“Not true at all”, “Rarely true”, “Sometimes true”, “Often true”, and “True nearly all of the time”). From these individual CDRISC item ratings, a total score of between 0 and 100 is calculated, with higher scores representing greater resilience. CDRISC total scores are significantly correlated (0.83) with total scores on the Kobasa Hardiness Measure and negatively correlated with total scores on the Perceived Stress Scale (−0.76). Internal consistency is good (Cronbach’s alpha = 0.89) and test–retest reliability (r = 0.87) is satisfactory [15]. According to the authors of the CDRISC, it is composed of five factors, with items loading most strongly on these factors shown in Table 1. For the purposes of data analysis, these factors are referred to as “RF1”, “RF2”, etc., below.

The 20-item Self-rating Depression Scale (SDS [33]) includes the diagnostic criteria and several associated features of the most recent definition of Major Depressive Disorder [2]. Respondents are invited to indicate the frequency of each of the 20 SDS items for them during the last two weeks by answering “None or a little of the time” (scored as (1)), “Some of the time” (2), “Good part of the time” (3), or “Most or all of the time” (4), providing total raw scores from 20 to 80 (used in this study). SDS raw scores of 40 or above indicate the presence of “clinically significant depression” [34]. The SDS has demonstrated a split-half reliability of 0.81 [33], 0.79 [35], and 0.94 [36], and an internal consistency (alpha) of 0.88 for depressed patients and 0.93 for non-depressed patients [37]. SDS total score was used to classify participants into “depressed” versus “non-depressed” on the basis of Zung’s cutoff score of 40.

EEG was recorded using a 40-channel Neuroscan QuikCap (Compumedics USA Ltd., El Paso, TX, USA) with electrodes arranged in accordance with the international 10/20 system and aligned with the anatomical nasion and inion points. Electrodes were composed of sintered Ag/AgCl. Signals were acquired and digitised using a NuAmps digital amplifier (Compumedics USA Ltd., El Paso, TX, USA) at a sampling rate of 1000 Hz and passed through a bandpass filter of DC to 250 Hz. The amplifier was connected to Curry 7 Acquisition software (Compumedics USA Ltd., El Paso, TX, USA) running on a Dell Optiflex 9020 desktop PC (Dell, Brisbane, QLD, Australia). Recordings were referenced to the average of the A1-A2 earlobe electrodes and later converted to a common average reference offline. EOG data were collected using four electrodes: two arranged above and below the left eye to measure vertical eye movement and two more arranged at outside the left and right canthus to measure horizontal eye movement. Impedance values at all electrodes were <5 kΩ prior to the start of recording.

Participants read an explanatory statement and were given the opportunity to ask any questions before giving written consent to participate. Participants completed a background questionnaire (age, sex), the SDS, and the CD-RISC, after which the electrode cap was fitted. Participants were then seated in the experimental booth to minimise external stimuli, had headphones placed upon their ears, and were asked to relax. After 15 min of sitting still (adaptation), the audio-recorded experimental protocol (3 min Eyes Open, 3 min Eyes Closed) was presented via headphones to ensure consistency across participants. Following the end of the protocol, all equipment was removed from the participant, who was thanked for their participation. Ethics approval for this study was provided by the Human Research Ethics Committee of the University of New England, Australia (Approval No. HE14-051), consistent with the Code of Ethics of the World Medical Association (Declaration of Helsinki). This research did not receive any specific grant from funding agencies in the public, commercial, or not-for-profit sectors.

Data were processed using a 1–45 Hz 2nd order Butterworth bandpass filter and then (as mentioned above) re-referenced to a common average. Data tapering was conducted using a Hann window with a 10% width to prevent data loss. EEG data were visually examined to identify artefacts (eye movements, muscle movements, spontaneous discharges or electrode pops, etc.), which were then removed from the data record. Bad block and eye blink detection (using the magnitude of eye blink deflections as a set threshold criterion to detect artefacts) was undertaken by three automated methods (Subtraction, Covariance and Principal Component Analysis) to produce clean EEG data.

Back-to-back epochs of 2 s duration were then created from the cleaned EEG data. Epochs with bad blocks were excluded from the averaged data. Most participants had over 90% usable artefact-free epochs for both the Eyes Open and Eyes Closed conditions, with the lowest frequencies of such usable epochs being 87% and 49% for the Eyes Open and Eyes Closed conditions, respectively.

Functional lagged linear connectivity (also known as. coherence) estimates of EEG frequency band activity were obtained for theta (4–8 Hz), alpha (8–12 Hz), and beta (12–30 Hz) for each available epoch using The Key Institute eLORETA (exact low-resolution brain electromagnetic tomography [29]) software. This technique provides a single weighted minimum norm solution to the inverse problem and has been demonstrated to provide zero error (but low spatial resolution) in localising cortical grey matter test sources [38,39]. The weights utilised by eLORETA yield images of current source density in a standardised realistic head model (Fuchs et al., 2002) based on the MNI152 template [40]. Regions of interest (ROIs) were selected using commonly identified grey matter nodes in the DMN, SN, and CEN based on MNI coordinates as identified by Raichle [41], with all grey matter tissue within 10 mm of the identified source included as part of that node. This resulted in 18 ROIs being selected (see Table 2).

## 3. Results

### 3.1. Data

Table 3 shows the descriptive data for the CDRISC in both depressed and non-depressed groups. Some aspects of these data have been published previously [42]. The 5% trimmed means were very close to the actual means, suggesting that there were negligible effects from outliers, although skewness was present. There was no significant correlation between the sex of participants and CDRISC total score (ρ = 0.091, *p* = 0.366) or SDS total score (ρ = 0.022, *p* = 0.829), or between the age of participants and CDRISC (ρ = 0.060, *p* = 0.553) or SDS (ρ = 0.055, *p* = 0.584), allowing the data to be analysed without adjusting for those potential confounds. Independent-samples *t*-tests showed that the depressed group displayed consistently lower RF scores compared to the non-depressed group, except for RF 5, which showed no significant difference between the groups (see Table 4).

### 3.2. Alpha Band

Significant positive correlations were found between alpha band coherence and RF3 in the depressed group. As RF3 scores increased, alpha functional connectivity increased between the left inferior temporal lobule (DMN) and two other nodes: the right lateral parietal lobule (DMN; r = 0.498, *p* = 0.065) and the right superior parietal lobule (CEN; *r* = 0.492, *p* = 0.071, see Figure 1). No other significant correlations were found in the alpha band across any resilience factor.

### 3.3. Beta Band

Significant negative correlations were found between beta band coherence and RF1 in the depressed group. As RF1 scores decreased, beta functional connectivity also increased between the left superior parietal lobule (CEN) and the left insula (SN; *r* = −0.535, *p* = 0.012, see Figure 2).

The depressed group also showed significant negative correlations between beta band coherence and RF3. As RF3 scores decreased, beta functional connectivity increased between the left superior parietal lobule (CEN) and two other nodes: the dorsal medial prefrontal cortex (CEN; *r* = −0.502, *p* = 0.081) and the left insula (SN; *r* = −0.498, *p* = 0.088, see Figure 3).

Significant positive correlations were found between beta band coherence and RF4 in the non-depressed group. As RF4 scores increased, beta functional connectivity increased between the posterior cingulate (DMN) and three other nodes: the right lateral parietal lobule (DMN; *r* = 0.316, *p* = 0.096), the left inferior temporal lobule (DMN; *r* = 0.375, *p* = 0.008), and the left superior parietal lobule (CEN; *r* = 0.357, *p* = 0.018, see Figure 4).

The non-depressed group also showed significant positive correlations between beta band coherence and RF5. As RF5 scores increased, beta functional connectivity increased between the left superior parietal lobule (CEN) and two other nodes: the posterior cingulate (DMN; *r* = 0.359, *p* = 0.033) and the left lateral parietal lobule (DMN; *r* = 0.471, *p* < 0.001, see Figure 5).

### 3.4. Theta Band

No significant results across any RFs were found in the theta band in either the depressed or non-depressed group.

## 4. Discussion

This study examined the correlations between depression-related resilience factors (as derived from the CDRISC) and eyes-closed resting-state theta/alpha/beta coherence between multiple nodes of three major neural networks, as well as how these correlations varied between a clinically depressed group and a non-depressed control group. Whilst no significant differences were found in theta band coherence, the depressed group showed a positive correlation between RF3 scores and alpha coherence within the DMN and a connection between the DMN and CEN. Multiple yet consistent differences were found in beta coherence, with the depressed group showing negative correlations between RF1/3 scores and beta coherence within the CEN and between the SN and the CEN. This contrasted with the non-depressed group, who showed positive correlations between RF4/5 scores and beta coherence within the DMN and between the DMN and CEN.

### 4.1. Theta

The evidence from previous studies regarding theta functional connectivity in depressed populations is currently mixed. Whilst several studies [43,44] reported that depressed people typically show lower theta connectivity compared to non-depressed people, the opposite (an increase in theta connectivity) is also reported in several studies [45,46]. As such, no hypothesis regarding theta connectivity could be made in advance with any degree of confidence. The absence of positive findings in this study indicates that theta activity may not indicate fragility in those with MDD.

### 4.2. Alpha

As reported above, the depressed group showed a positive correlation between RF3 (confidence in security of relationships) and alpha band coherence, primarily in the DMN. It is commonly reported that increased alpha activity in the DMN negatively affects task performance [47,48,49]. If an increase in DMN alpha connectivity is associated with irregular mental function, it follows that we would expect some differences in alpha connectivity in the depressed sample. Indeed, Fingelkurts and Fingelkurts [23] reported an increase in alpha synchronisation between multiple nodes of the DMN during resting state.

That said, the positive correlation between alpha coherence and RF3 indicates that an increase in alpha connectivity was associated with greater confidence in accepting change and the security of personal relationships for depressed people. This would appear to be an example of alpha inhibition as an adaptive response to minimise maladaptive thoughts. Alpha connectivity is believed to play a major role in information processing as a gating mechanism by inhibiting brain regions that are irrelevant to the matter at hand [50]. In this case, those in the depressed group with greater confidence in accepting change and the security of their personal relationships appear to require greater alpha connectivity in order to inhibit parts of the DMN that would normally engage in dysfunctional thoughts, such as ruminating over previous events or negative self-talk.

### 4.3. Beta

As with theta connectivity, previous reports show evidence in both directions regarding beta connectivity and depression, with multiple reports of comparatively high beta connectivity in depressed groups [32,51], comparatively low beta connectivity (e.g., [52,53]), and no differences in beta connectivity [54,55]. This re-emphasizes the view that depression is not a unitary disorder but is rather a broad set of maladaptive emotional states that requires a greater understanding of the subtypes of depression and how they are expressed in neural function (or, in this case, dysfunction).

### 4.4. Depression-Based Differences

Consistent distinctions were found here between the depressed and non-depressed groups in terms of beta coherence for all statistically significant results: the depressed group showed negative correlations between beta coherence and RF scores, while the non-depressed group showed positive correlations. The significant connections in the depressed group were all either within the CEN or between the CEN and the left insula (SN), whilst the connections for the non-depressed group were either within the DMN or between the DMN and the left superior parietal lobule (CEN), bypassing the SN altogether. The left superior parietal lobule played a role in all four significant correlations regarding beta connectivity (both negative and positive), indicating that this structure may be important in emotional resilience.

### 4.5. The Depressed Group

As listed above (see Results), the depressed group showed negative correlations between beta coherence and two factor scores: RF1 (notion of personal competence, high standards, and tenacity) and RF3 (positive acceptance of change and secure relationships). Both connectivity results involved connections between the left insula and the left superior parietal lobule. There are several possible interpretations for these results. While normal levels of beta activation are often associated with cognitive concentration, it would not be expected in an eyes-closed resting-state condition (as used in this study). Excess beta power can be seen as a sign of drowsiness and/or sedation, particularly in regions related to sensorimotor organisation and orientation [56]. This would certainly apply to the left superior parietal lobule, which is highly involved in planned movements, spatial reasoning/orientation, and attention. An excess of beta power may result in top-down beta suppression of the insula by the left superior parietal lobule, resulting in a lack of motivation and affect that would be expressed in a lack of self-confidence (RF1) and/or a sense of insecurity in their relationships (RF3).

Alternately, these results may be a product of bottom-up processing from the left insula. The insula is involved in a wide range of emotional processing tasks as part of the SN, including awareness of emotional state, identifying social norm violations, and maintaining general emotional regulation [57,58]. Rather than an excess of beta resulting in suppressed emotional regulation, these results may be a product of the insula performing its normal emotional regulatory function but in an undesired way. While the insula is key in maintaining awareness of emotional state and ongoing emotional regulation, this does not mean it correctly identifies or maintains the current and appropriate emotional state. Since all the negative correlations in these results were found in the depressed group, and the insula is only implicated in those negative correlations and no other results, the insula may have a key role in these two specific RFs for those with depression.

### 4.6. The Non-Depressed Group

As listed above, the non-depressed group showed positive correlations between beta coherence and two factor scores: RF4 (personal control) and RF5 (spiritual influences). Similar to the significant results from the depressed group, each of these results involved connectivity with the left superior parietal lobule. Additionally, rather than the left insula, both significant beta coherence correlations in the non-depressed group also involved the posterior cingulate, widely considered the key node of the DMN [59].

Distinct beta activation patterns have been observed within the DMN, and the posterior cingulate has been observed to produce strong beta connectivity with functional cortical structures outside the DMN during resting states [60,61]. Whilst it is common for inter-network connections between the DMN and CEN to be mediated by the SN, the positive correlations found here bypass the SN. This contrasts with the negative correlations in the depressed group, which occur between the SN and CEN and do not involve the DMN at all. One interpretation of the overall beta connectivity results is that unhealthy emotional resilience may be a product of unnecessary increased beta activation in the SN (particularly the left insula), in which the insula may be providing unreliable information to the CEN regarding what stimuli and emotions are salient to the situation.

It should be emphasised that this study’s findings do not contradict any previous report on functional connectivity in depression, as that was not the purpose of this experiment. The focus of this study was to examine how emotional resilience can be seen in resting-state EEG and how these connectivity measures differ in those with depression. These findings are not intended to provide commentary on general differences in functional connectivity due to depression but instead to explore how differences in emotional resilience can affect the baseline activity in the brain in those with depression compared to the general population.

### 4.7. Limitations

The comparison of “depressed” and “non depressed” subsamples was based upon the criteria set out by Zung [33] and, although this has been confirmed in previous validity studies [35], the addition of a clinician interview to the assessment of depression would help ascertain the reliability of that process in future studies on PR and EEG data in community samples. The CDRISC is well established in the literature on PR but there are other measures that might include additional factors than the five in the CDRISC, thus potentially enhancing the generalizability of these findings. Similarly, due to the “snapshot” nature of the current study, no inferences can be drawn regarding how PR is associated with specific EEG data over time, during periods of stress, or as a result of specific events. The sample consisted of volunteers from one specific cultural and geographical setting, and generalisation to other places and to non-volunteers could enhance these findings. EEG data are well established as representation of neurocognitive activity, but other methodologies for investigating brain activity (e.g., fMRI) might reveal alternative associations with PR. These limitations must be acknowledged but do not invalidate the findings reported here.

### 4.8. Clinical Implications

Although traditional therapeutic models based upon medication or verbal psychotherapy have been demonstrated to influence functional connectivity [62,63,64,65], and therefore remain relevant in the light of these findings, the value of more direct neurotherapies for depressed persons, including various forms of neurofeedback, is highlighted here. For example, fMRI-based neurofeedback has been shown to reverse abnormal hypoconnectivity to the amygdala [66], and depressive symptoms have been reduced following normalisation of the dorsolateral prefrontal cortex-precuneus connectivity after functional connectivity neurofeedback [67]. However, the particular contribution that this research makes to depression therapies is via the link between brain connectivity, psychological fragility, and depression. Psychological resilience can be taught [68,69], and the role of neurofeedback in resilience training by focusing upon enhancing specific functional connectivity between brain sites and within selected wavelengths has received some attention [70,71]. The translation of these research findings to the clinical setting remains relatively under-established, and the results of the current study help to provide some focus for those endeavours in terms of brain connections and wavelengths of brain electrical activity. By focusing (for example) upon beta coherence within the CEN, SN, and DMN (mentioned above in Section 4.4), neurofeedback therapies might bridge the current gap between more generalised connectivity neurofeedback and the application of that treatment to the enhancement of psychological resilience for depressed patients.

## 5. Conclusions

The consistent presence of the left parietal lobule in these beta coherence results indicates that it is an important structure in terms of psychological fragility. The positive beta coherence correlations between RFs and the left parietal lobule in the non-depressed group, along with the negative correlations found between other RFs at the same site in the depressed group, indicate that the left lateral parietal lobule may play an important role in emotional regulation.

## Figures and Tables

**Figure 1 brainsci-14-00845-f001:**
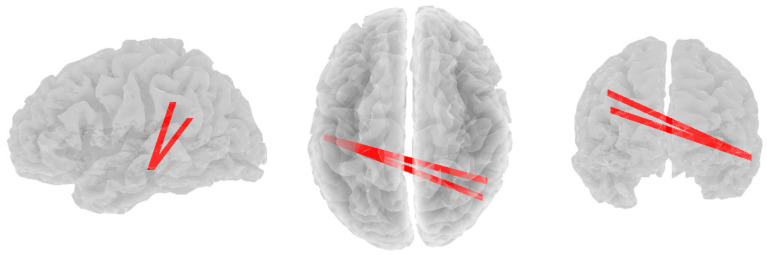
Significant positive correlations between alpha band coherence and RF3 in the depressed group. Views are from the left, top, and rear, respectively.

**Figure 2 brainsci-14-00845-f002:**
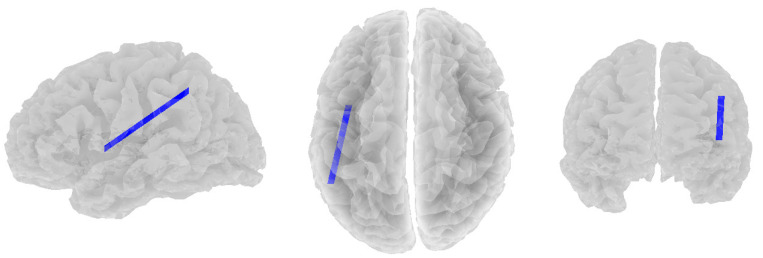
Significant negative correlations between beta band coherence and RF1 in the depressed group. Views are from the left, top, and rear, respectively.

**Figure 3 brainsci-14-00845-f003:**
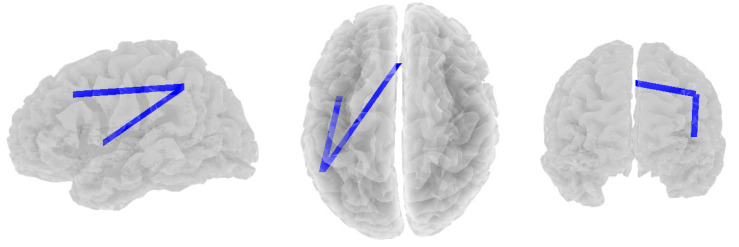
Significant negative correlations between beta band coherence and RF3 in the depressed group. Views are from the left, top, and rear, respectively.

**Figure 4 brainsci-14-00845-f004:**
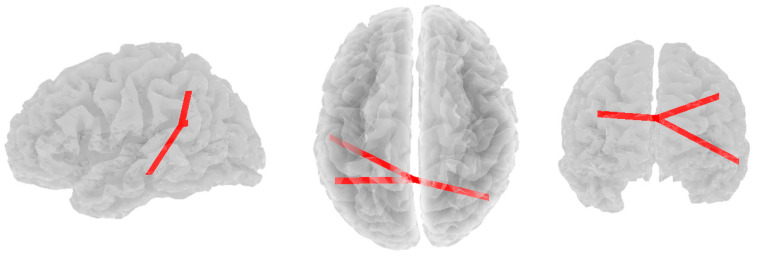
Significant positive correlations between beta band coherence and RF4 in the non-depressed group. Views are from the left, top, and rear, respectively.

**Figure 5 brainsci-14-00845-f005:**
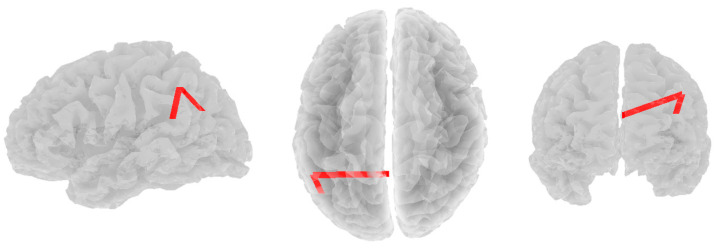
Significant positive correlations between beta band coherence and RF5 in the non-depressed group. Views are from the left, top, and rear, respectively.

**Table 1 brainsci-14-00845-t001:** Connor–Davidson Resilience Scale Factors and items.

CDRISC Factors	CDRISC Items
1: Personal competence, high standards, and tenacity	10. I give my best effort no matter what
	11. I can achieve my goals
	12. When things look hopeless, I don’t give up
	16. I am not easily discouraged by failure
	17. I think of myself as a strong person
	23. I like challenges
	24. I work to attain my goals
	25. I have pride in my achievements
2: Trust in one’s instincts, tolerance of negative affect, and strengthening effects of stress	6. I can see the humorous side of things
	7. I believe that coping with stress strengthens me
	14. When I’m under pressure, I can focus and think clearly
	15. I prefer to take the lead in problem solving
	18. I make unpopular or difficult decisions
	19. I can handle unpleasant feelings
	20. I have to act on a hunch
3: Positive acceptance of change and secure relationships	1. I am able to adapt to change
	2. I have close and secure relationships
	4. I can deal with whatever comes
	5. Past success gives me confidence for new challenges
	8. I tend to bounce back after illness or hardship
4: Control	13. I know where to turn for help
	21. I have a strong sense of purpose
	22. I am in control of my life
5: Spiritual Influences	3. Sometimes fate or God can help me
	9. Things happen for a reason

**Table 2 brainsci-14-00845-t002:** Neural networks, brain sites, and MNI coordinates for central voxel of each ROI.

Network	Location	MNI (X, Y, Z)
Default Mode Network	posterior cingulate	0, −52, 27
	medial prefrontal cortex	−1, 54, 27
	left lateral parietal lobule	−46, −66, 30
	right lateral parietal lobule	49, −63, 33
	left inferior temporal lobule	−61, −24, −9
	right inferior temporal lobule	58, −24, −9
Central Executive Network	dorsal medial prefrontal cortex	0, 24, 46
	left anterior prefrontal cortex	−44, 45, 0
	right anterior prefrontal cortex	44, 45, 0
	left superior parietal lobule	−50, −51, 45
	right superior parietal lobule	50, −51, 45
Salience Network	dorsal anterior cingulate	0, −21, 36
	left anterior prefrontal cortex	−35, 45, 30
	right anterior prefrontal cortex	32, 45, 30
	left insula	−41, 3, 6
	right insula	41, 3, 6
	left lateral parietal lobule	−62, −45, 30
	right lateral parietal lobule	62, −45, 30

Locations and coordinates derived from Raichle, 2011 [41].

**Table 3 brainsci-14-00845-t003:** Descriptive data for CDRISC in both depressed (D) and non-depressed (ND) groups.

Resilience Factor	Group	Mean	SD	SEM	5% Trimmed Mean
1	ND	4.18	0.51	0.06	4.20
	D	3.28	0.66	0.12	3.27
2	ND	3.63	0.39	0.05	3.64
	D	2.95	0.55	0.10	2.95
3	ND	4.36	0.49	0.06	4.38
	D	3.25	0.75	0.13	3.25
4	ND	4.09	0.67	0.08	4.13
	D	3.10	0.88	0.15	3.09
5	ND	2.13	0.80	0.10	2.15
	D	2.05	0.84	0.15	2.06

**Table 4 brainsci-14-00845-t004:** Independent-samples *t*-test for differences between depressed and non-depressed groups. Variances between groups was approximately equal across all tests (df = 98).

Resilience Factor	Group Difference	95% CI	*t*	*p*	Cohen’s *d*
1	0.899	0.66–1.14	7.4520	<0.001	1.59
2	0.681	0.49–0.87	7.1460	<0.001	1.52
3	1.11	0.86–1.36	8.8370	<0.001	1.88
4	0.988	0.67–1.30	6.2350	<0.001	1.33
5	0.084	−0.26–0.43	0.4830	0.63	0.1

## Data Availability

The raw data supporting the conclusions of this article will be made available by the authors on request.

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
