# Peer review of "Functional Network Connectivity for Components of Depression-Related Psychological Fragility"

_brainsci, 2024, doi:10.3390/brainsci14080845_

Round 1
Reviewer 1 Report
Comments and Suggestions for Authors
The authors performed a very interesting and extensive work, but for their manuscript improvement, I recommend the following adjustments or added information:
Material and methods
Please provide power and sample size calculations for the analyzed population.
https://www.ncbi.nlm.nih.gov/pmc/articles/PMC7745163/
In Table 3, I suggest that the authors provide a mean of total scores for each questionnaire dimension rather than a mean for each answer (for example, the resilience factor from Table 1 has 8 questions; therefore, the sum of the answers at the highest level could be 320, especially since a higher score on CDRS = higher resilience). Furthermore, the new data can be more efficient for further statistical analysis (for example, linear regression).
The authors should consider performing a simple or multiple linear regression. This analysis can be a powerful tool to identify associations and predictions between variables, potentially providing more insightful information. It would be particularly interesting to see if any dimension/factor of both questionnaires can be associated with different band coherence ( Theta, Alpha, Beta, Gama).
https://www.ncbi.nlm.nih.gov/pmc/articles/PMC5122272/
Author Response
Reviewer comments: Please provide power and sample size calculations for the analyzed population
Author response: A good suggestion; this information is now included in the first paragraph of the Materials and Methods section: "Statistical power analysis indicated that a minimum sample of 71 was required, assuming α- error probability of 0.1, power of 0.9 and H1 correlations of at least 0.3."
Reviewer comments: In Table 3, I suggest that the authors provide a mean of total scores for each questionnaire dimension rather than a mean for each answer (for example, the resilience factor from Table 1 has 8 questions; therefore, the sum of the answers at the highest level could be 320, especially since a higher score on CDRS = higher resilience). Furthermore, the new data can be more efficient for further statistical analysis (for example, linear regression).
Author responses: Whilst this linear transformation is possible, presenting the results in this fashion would give the reader the false impression that some resilience factors have greater weightings than other factors solely based on the number of questions behind them.
Reviewer comments: The authors should consider performing a simple or multiple linear regression. This analysis can be a powerful tool to identify associations and predictions between variables, potentially providing more insightful information. It would be particularly interesting to see if any dimension/factor of both questionnaires can be associated with different band coherence ( Theta, Alpha, Beta, Gama).
Author responses:
We agree that such analyses would be quite enlightening, however the EEG functional connectivity technique used (eLORETA) is not capable of such statistical analyses. Whilst eLORETA can calculate beta coefficients for a single regressor (and the relevant constant), it does not report the relevant p-values that would have to be reported. This is likely the result of a range of the complex mathematics that eLORETA has to overcome in order to solve the inverse problem via nonparametric permutation testing (see Nichols & Holmes, 2002; https://doi.org/10.1002/hbm.1058).
Reviewer 2 Report
Comments and Suggestions for Authors
The manuscript is well-crafted and the presentation of the results is of value. The study examines the relationship between psychological resilience and functional connectivity in the alpha and beta bands within (and between) eighteen established cortical nodes in the default mode network, the central executive network, and the salience network. The results indicate that depression may be modulating how resilience is expressed in terms of fundamental neural activity.
Some concerns: Please clearly state what was the level of resilience among the participants, how many of them have high resilience and how many low resilience.
Author Response
Reviewer comments: Some concerns: Please clearly state what was the level of resilience among the participants, how many of them have high resilience and how many low resilience.
Author response: Certainly; data related to resilience factor scores for both the depressed and non-depressed groups can be found in Table 3. The study did not use any form of high/low categorization for RF scores as part of the analysis, but instead measured the correlation between RF scores and functional coherence. This required resilience levels to be measured as continuous variables. The only high/low categories used were those for high and low depression scores based on the Self-rating Depression Scale (see page 4).
Reviewer 3 Report
Comments and Suggestions for Authors
As psychological resilience is known to be inversely associated with depression, in the present investigational study the Authors evaluated the relationship between psychological resilience and functional connectivity in the alpha and beta bands within (and between) eighteen established cortical nodes in the default mode network, the central executive network, and the salience network (that are crucial in major depression).
Overall, I found the present study timely, original, and scientifically sound. Nevertheless, I have some major comments aimed at improving the quality of the paper, and these are outlined below.
- In the introduction a brief note on the fact that major depressive disorder has several subtypes that might underpin different neurobiological pathways and brain circuits, should be added with appropriate references (please, see doi 10.1080/09540261.2020.1765517).
- All subjects included in the study were recruited from a media-advertised study to “investigate how you feel”. This is too vague and, please, be more precise and report important information on study sample. For example, were the participants provided with a recompense?
- Besides, how many subjects were screened, but not included?
- Was also the presence of an intellectual disability assessed and how?
- Translating into “real world” clinical practice and medicine, what possible clinical shreds of evidence might arise from the present study and what the Researchers do suggest to improve interventions? Please add a brief paragraph (or a box) on “recommendations” or “clinical keypoints” in terms of integrative care.
Author Response
Reviewer comments: In the introduction a brief note on the fact that major depressive disorder has several subtypes that might underpin different neurobiological pathways and brain circuits, should be added with appropriate references (please, see doi 10.1080/09540261.2020.1765517).
Author responses: This information has now been included in the extended first paragraph of the introduction:
"These combinations have been referred to as ‘depression subtypes’, and several of these subtypes have been defined. For example, the DSM-5-TR [2] lists MDD with anxiety, with mixed features, with melancholic features, and with atypical features, plus others. Other MDD subtypes have been described by various authors [3-6], including the possible different neurobiological pathways and circuits they may have."
Reviewer comments: All subjects included in the study were recruited from a media-advertised study to “investigate how you feel”. This is too vague and, please, be more precise and report important information on study sample. For example, were the participants provided with a recompense?Besides, how many subjects were screened, but not included?
Author responses: No payment or recompense was provided to participants. Six subjects failed to meet the inclusion criteria, and a further subject was excluded due to insufficient epochs for the EEG analysis. This information, as well as the updated description of the study sample is also now included in the first paragraph of the Materials and Methods section:
"A total sample of 106 adults aged 18yr or more from the New England region of New South Wales, Australia, were recruited from a media-advertised study investigating mental health in rural and regional communities of Australia as part of the New England Mental Health Study. Participants were screened for: no previous medical history of severe physical brain injury, previous brain surgery, or past or current history of epilepsy or seizure disorder, claustrophobia (EEG data were collected in a small booth) or undergoing pharmacotherapy at the time of data collection. Six participants were excluded based on these criteria. One further participant was excluded from the EEG analysis due to an exceptionally low number of artifact-free epochs in the baseline condition, resulting in a final sample of 99 (46 females)."
Reviewer comments: Was also the presence of an intellectual disability assessed and how?
Author responses: No; not only would this have violated the requirements of the university's ethics committee, the additional time required to assess (with a satisfactory degree of reliability) any of the wide range of potential intellectual disabilities would have extended the testing time to a point where boredom effects would contaminate the alpha and beta band results of the EEG recording (see Katayama & Natsume, 2012: https://doi.org/10.2299/jsp.16.637).
Reviewer comments: Translating into “real world” clinical practice and medicine, what possible clinical shreds of evidence might arise from the present study and what the Researchers do suggest to improve interventions? Please add a brief paragraph (or a box) on “recommendations” or “clinical keypoints” in terms of integrative care.
Author comments: This information is now included in the final paragraph of the Discussion section under 4.8: Clinical Applications:
“4.8. Clinical Implications
Although traditional therapeutic models based upon medication or verbal psychotherapy have been demonstrated to influence functional connectivity [62-65], and therefore remain relevant in the light of these findings, the value of more direct neuro-therapies for depressed persons, including various forms of neurofeedback, is high-lighted here. For example, fMRI-based neurofeedback has been shown to reverse ab-normal hypoconnectivity to the amygdala [66], and depressive symptoms have been reduced following normalisation of the dorsolateral prefrontal cortex-precuneus connectivity after functional connectivity neurofeedback [67]. However, the particular contribution that this research makes to depression therapies is via the link between brain connectivity, psychological fragility, and depression. Psychological resilience can be taught [68, 69], and the role of neurofeedback in resilience training by focusing upon enhancing specific functional connectivity between brain sites and within selected wavelengths has received some attention [70, 71]. The translation of these research findings to the clinical setting remains relatively under-established, and the results of the current study help to provide some focus for those endeavours in terms of brain connections, and wavelengths of brain electrical activity. By focusing (for example) upon beta coherence within the CEN, SN and DMN (mentioned above in 4.4), neurofeedback therapies might bridge the current gap between more generalised connectivity neurofeedback, and application of that treatment to the enhancement of psychological resilience for depressed patients.”
Round 2
Reviewer 1 Report
Comments and Suggestions for Authors
The authors performed the recommended adjustments. Therefore, the paper is suitable for publication.